# SOUL: Towards Sentiment and Opinion Understanding of Language

**Yue Deng**[* 1,2]  **Wenxuan Zhang**[† 1,3]  **Sinno Jialin Pan**[2,4]  **Lidong Bing**[1,3]

[1]DAMO Academy, Alibaba Group, Singapore  [2]Nanyang Technological University, Singapore
[3]Hupan Lab, 310023, Hangzhou, China  [4]The Chinese University of Hong Kong
{yue.deng, saike.zwx, l.bing}@alibaba-inc.com
sinnopan@cuhk.edu.hk

## Abstract

Sentiment analysis is a well-established natural language processing task, with sentiment polarity classification being one of its most popular and representative tasks. However, despite the success of pre-trained language models in this area, they often fall short of capturing the broader complexities of sentiment analysis. To address this issue, we propose a new task called Sentiment and Opinion Understanding of Language (SOUL). SOUL aims to evaluate sentiment understanding through two subtasks: Review Comprehension (RC) and Justification Generation (JG). RC seeks to validate statements that focus on subjective information based on a review text, while JG requires models to provide explanations for their sentiment predictions. To enable comprehensive evaluation, we annotate a new dataset comprising 15,028 statements from 3,638 reviews. Experimental results indicate that SOUL is a challenging task for both small and large language models, with a performance gap of up to 27% when compared to human performance. Furthermore, evaluations conducted with both human experts and GPT-4 highlight the limitations of the small language model in generating reasoning-based justifications. These findings underscore the challenging nature of the SOUL task for existing models, emphasizing the need for further advancements in sentiment analysis to address its complexities. The new dataset and code are available at https://github.com/DAMO-NLP-SG/SOUL.

## 1 Introduction

Sentiment analysis, a well-established natural language processing task, aims to analyze and understand subjective information from text (Liu, 2015). One of its most popular and representative tasks is sentiment classification (SC), which involves

Figure 1: Examples from SOUL. We also show traditional sentiment classification task for comparison.

classifying a given text like customer review to a pre-defined sentiment label, such as positive, negative, or neutral (Pang and Lee, 2005; Maas et al., 2011; Socher et al., 2013; Zhang et al., 2015). With the advent of pre-trained language models, especially the recent large language models (LLMs), remarkable performance has been achieved on SC which sometimes even surpasses human performance (Yin et al., 2020; Ke et al., 2020; Zhang et al., 2021; Fan et al., 2022; Wang et al., 2023; Zhang et al., 2023). This leads to a common belief that SC, and sentiment analysis in general, has reached its saturation.

However, SC is not equivalent to the broader field of sentiment analysis as it does not require a deep understanding of the underlying sentiments and opinions expressed in the text. To determine the overall sentiment orientation, a model can simply rely on superficial textual features, such as the presence of specific words or phrases indicating positivity or negativity (Wulczyn et al., 2017; Wang and Culotta, 2020, 2021; Moon et al., 2021; Choi et al., 2022). Therefore, even if a model demon-

---

[*] Yue Deng is under the Joint PhD Program between Alibaba and Nanyang Technological University.

[†] Wenxuan Zhang is the corresponding author.

strates satisfactory performance in sentiment classification, it may not fully capture the subtle nuances of sentiment in languages, such as mixed sentiments towards different aspects, motivation of the expressed opinions, and possible outcomes of such sentiments, etc. In order to assess whether a model can truly comprehend the sentiment and accurately interpret intricate emotions, it is essential to adopt a more comprehensive approach that extends beyond merely predicting the polarity of sentiment.

To this end, we introduce a new sentiment analysis task, namely Sentiment and Opinion Understanding of Language (SOUL). Our inspiration comes from reading comprehension tasks, which assess human understanding of a passage by asking to judge the validity of a statement. Similarly, we adopt the form of verifying comprehension statements regarding an opinionated review text. We also generate justifications for such predictions as a means of testing the sentiment understanding capability of models. As shown in Figure 1, given a review text, as well as statements that focus on subjective information discussed in the review, SOUL features two novel subtasks: Review Comprehension (RC) and Justification Generation (JG).

Specifically, the RC task aims to determine if the given statement is `true`, `false`, or `not-given` based on the review, answering the question of *what* the sentiment is. While this task still involves a classification format, it can cover a broad range of sentiment phenomena with the flexibility to create statements focusing on diverse subjective aspects of the text. This flexibility breaks the restriction of SC purely focusing on sentiment polarity and allows for the introduction of more complex sentiment problems. In Figure 1, the reviewer's sentiment towards the raptor graphics lacks specific reasons, making it difficult for a simple pattern matching model to accurately predict the first statement as `not-given` without contextual understanding. The second statement in Figure 1 also presents a challenge for models in detecting sarcasm. The JG task, on the other hand, seeks to provide an explanation for the rationale behind the model's interpretation of sentiment, answering the question of *why* the sentiment is as predicted. By generating justifications for its predicted label, the model is forced to consider the context and nuances of the input text, rather than relying solely on superficial features such as individual words or phrases. For example, the second justification in Figure 1 explains why

the statement is `false` and identifies the sarcastic meaning conveyed by the reviews.

To facilitate such an investigation, we carefully annotate a new dataset based on common review corpora. In total, it consists of 15,028 statements across 3,638 reviews. Each statement is also annotated with a label and the corresponding justification. We extensively benchmark SOUL with both small language models (SLMs) trained with the complete training set and also LLMs under the zero-shot setting. Our experimental results indicate that SOUL is a challenging task that demands a deep understanding of sentiment, with a performance gap of up to 27% when compared to human performance. In addition, based on comprehensive evaluations conducted by both human experts and the GPT-4 model, it has been observed that SLMs have demonstrated proficiency in validating statements but struggle with generating reasoning-based justifications, indicating significant potential for enhancement in their comprehension of sentiment. In comparison, ChatGPT's strength lies in producing well-reasoned justifications, showcasing its powerful sentiment-understanding ability. However, there is still room for improvement regarding the overall accuracy, originality, and conciseness of ChatGPT's responses. Overall, we believe SOUL will advance sentiment analysis and encourage the creation of models capable of understanding sentiments at a human-like proficiency.

## 2 SOUL

### 2.1 Task Formulation

Let $t$ be an opinionated text item (e.g., a product review); $s$ be a textual statement about the subjective information in the text; $l \in \{\texttt{true}, \texttt{false}, \texttt{not-given}\}$ be the label of $s$; $j$ be the justification for $l$; $f$ be a model.

**Review Comprehension** The objective of RC is to determine the validity $l$ of the statement $s$ in relation to review $t$. This involves classifying the statement $s$ as either `true`, `false`, or `not-given`:

$$f(t, s) \rightarrow l \quad (1)$$

To accomplish this task effectively, a model must fully comprehend the subjective information presented in both the review and the statement, and subsequently judge the validity.

**Justification Generation** JG aims to generate predictions $l$ and justifications $j$ jointly:

$$f(t, s) \rightarrow l, j \qquad (2)$$

The purpose is to enable the model to generate a justification that explains its predicted label, thereby helping us to examine whether the model has truly understood the sentiment.

## 2.2 Dataset Construction

**Data Collection** We utilize review texts from two corpora: Yelp (Zhang et al., 2015) and IMDb (Maas et al., 2011). The Yelp dataset is a collection of business reviews from the Yelp website, while the IMDb corpus consists of movie and TV show reviews from the IMDb website. These two datasets cover various review types and are widely used in existing sentiment analysis research, e.g., classifying the sentiment polarity of a given review. Therefore, we also take them as our data source for constructing subjective statements.

**Statement and Justification Annotation** Firstly, we instruct annotators to propose several statements focusing on various subjective information given a review. To achieve this goal, we request annotators to focus on multiple crucial sentiment elements, including the sentiment of opinion, sentiment target, opinion holder, the reason for the opinion, customer intent, etc (Liu, 2015). Annotators are instructed to delve beyond the surface-level content and generate more challenging statements that require a deeper level of sentiment and opinion understanding ability. For instance, simply describing the user does not like the product is discouraged, but statements focusing on mixed sentiments towards various aspects, or the underlying reasons behind opinions are encouraged. In the meantime, the label of each statement is annotated. Unlike traditional natural language inference (NLI) tasks, the primary objective of statement annotation is to extract and label subjective information rather than establish logical connections or entailment between different texts. Besides, we request annotators to provide justifications for their proposed statements. These justifications provide the rationale behind the statement categorization. By treating them as the target in the JG task, we can gain valuable insight into the model's prediction processes and verify whether the model possesses real sentiment understanding ability.

**Data Validation and Processing** After the initial construction phase, a separate group of annotators classifies each proposed statement without access

| Split | # reviews | # statements | | | |
|---|---|---|---|---|---|
| | | True | False | Not-given | Total |
| Train | 2,182 | 3,675 | 3,000 | 2,159 | 8,834 |
| Dev | 365 | 617 | 503 | 361 | 1,481 |
| Test | 1,091 | 1,956 | 1,664 | 1,093 | 4,713 |
| Total | 3,638 | 6,248 | 5,167 | 3,613 | **15,028** |

Table 1: Statistics of SOUL dataset.

to the original labels, aiming to evaluate the quality of the constructed statements. In cases of conflicting classifications, an expert annotator is consulted to resolve the discrepancies and assign a final label. In addition, annotators are instructed to categorize statements as simple, medium, or hard to determine their difficulty level. Reviews containing only simple statements are excluded to maintain an appropriate level of challenge.

**Dataset Statistics** The SOUL dataset comprises 15,028 statements related to 3,638 reviews, resulting in an average of 4.13 statements per review. To create training, development, and test sets, we split the reviews in a ratio of 6:1:3, respectively. Detailed statistics can be found in Table 1.

## 3 Experiments

### 3.1 Setup

**Models** We benchmark SOUL with several widely used Small Language Models with the complete training set, including Roberta (Liu et al., 2019), T5 (Raffel et al., 2020), and Flan-T5 (Chung et al., 2022). We adopt the base version for each model type. In addition, we extend our analysis to two representative LLMs from the Flan and GPT model families, namely Flan-T5$_{XXL}$ (13B) (Raffel et al., 2020) and ChatGPT[1], respectively. We evaluate these LLMs under a zero-shot setting. To reduce variance, we report the average results with three random seeds. The detailed setup can be found in Appendix A.1.

**Evaluation Metrics** For the RC task, we report f1 scores for each class and the overall accuracy. For the JG task, we use different evaluation metrics for predictions $l$ and justifications $j$. We measure statement predictions $l$ using overall accuracy. For justifications $j$, we employ commonly used text generation metrics, including BLEU (Papineni et al., 2002), ROUGE(1/2/L) (Lin, 2004),

---
[1] We conducted the experiments using the May 24th version of ChatGPT.

| Models | $F1_{true}$ | $F1_{false}$ | $F1_{not\text{-}given}$ | Accuracy |
|---|---|---|---|---|
| *human* | 99.34 | 99.22 | 98.72 | 99.15 |
| Roberta | 78.21 | 76.19 | 69.29 | 75.49 |
| T5 | 81.04 | 78.22 | 71.23 | 77.87 |
| Flan-T5 | 82.01 | 80.27 | **72.64** | 79.28 |
| Flan-T5$_{XXL}$ | **87.82** | **82.34** | 69.81 | **81.72** |
| ChatGPT | 85.41 | 73.50 | 29.36 | 72.09 |

Table 2: Performance of review comprehension task. We report human performance as a reference.

| Models | Acc | Similarity Evaluation | | |
|---|---|---|---|---|
| | | BLEU | ROUGE(1/2/L) | BERTScore |
| T5 | 74.50 | 24.01 | 50.16/32.79/46.65 | 92.33 |
| Flan-T5 | 79.32 | **25.05** | **51.25/33.77/47.67** | **92.53** |
| Flan-T5$_{XXL}$ | **79.59** | 12.80 | 33.95/18.62/31.02 | 88.98 |
| ChatGPT | 78.04 | 11.32 | 39.64/20.79/33.79 | 90.16 |

Table 3: Performance of justification generation task.

and BERTScore (Zhang et al., 2020) to calculate their similarity with the annotated justifications.

## 3.2 Main Results

**Review Comprehension**   The results of the RC task are presented in Table 2. We can make the following observations: 1) All models exhibit limited sentiment ability, resulting in a performance gap of 17% to 27% compared to human performance. This shows the difficulty of the RC task, and there is still much room for improvement in developing models that can accurately comprehend sentiment and opinion. The challenges may arise from the complexity and diversity of statements that incorporate mixed sentiments, underlying reasons of opinions, and other aspects. 2) Among SLMs, Flan-T5 achieves the best performance, surpassing T5 with the same model size by 1.41%, possibly due to the effectiveness of instruction tuning during its training process. 3) LLMs demonstrate effective zero-shot ability, with Flan-T5$_{XXL}$ achieving the best results even without any training data. In particular, ChatGPT appears to have difficulty with the not-given class, due to its overconfidence to misclassify the not-given class as false. This failure shows the challenges posed by SOUL and emphasizes that a large model size alone is not sufficient to ensure comprehensive sentiment capabilities.

**Justification Generation**   We exclude Roberta from the JG task as it is a discriminative model and not well-suited for text generation tasks. The results for the JG task are presented in Table 3. We report commonly used text generation metrics as similarity evaluation and overall accuracy for reference. When it comes to accuracy, it appears that SLMs and Flan-T5$_{XXL}$ show either minimal improvement or even a negative impact. Instead, ChatGPT stands out with a notable improvement of approximately 6% in validating subjective statements, The incorporation of justifications likely

facilitated ChatGPT a more thorough comprehension of the sentiment conveyed, thereby enhancing its performance. However, this may require a strong reasoning ability, which is not observed in these SLMs. Therefore, attempting to perform justification generation could result in a decrease in accuracy performance. Regarding similarity evaluation, it can be inferred that Flan-T5 is capable of generating justifications that closely resemble the annotated justifications, whereas Flan-T5$_{XXL}$ exhibits the weakest performance in this respect. Nevertheless, the results obtained from the similarity evaluation contradict the overall accuracy, indicating a need for a more robust evaluation method.

## 3.3 Comprehensive Evaluation

There is a variation in accuracy between these two tasks, and there are conflicting evaluations of accuracy and similarity within the JG task. To perform a thorough analysis, we aim to assess the generated justifications using the following criteria, rated on a scale of 1 (poor) to 3 (excellent): 1) **Correct**: whether it is sensible and logical when compared to the gold label; 2) **Align**: whether it aligns with its generated label; 3) **Relevant**: whether it is relevant to the statement; 4) **Concise**: whether it is brief and concise; 5) **Original**: whether it demonstrates innovation and uniqueness. We sample 50 instances and utilize both human evaluators and the GPT-4 model (OpenAI, 2023) for assessment. See Appendix A.2 for the GPT-4 evaluation prompt.

The evaluation results are shown in Figure 2. We can see that while SLMs and Flan-T5$_{XXL}$ have satisfactory performance in the RC task, their justifications in the JG task lack originality, which means that they often rely on copied reviews without providing additional insights. This prediction process, without proper reasoning, potentially reduces its overall accuracy and creates inconsistencies between the two tasks. Conversely, ChatGPT exhibits promising performance across various criteria, indicating its robust sentiment understanding capability. Nevertheless, there is still room for im-

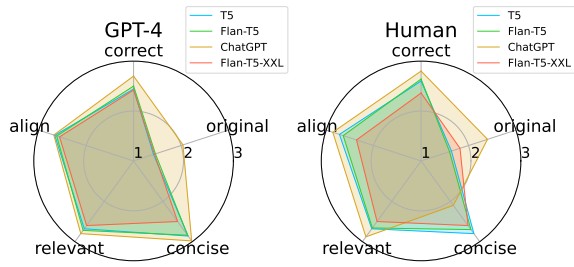

Figure 2: Evaluation results for generated justifications by GPT-4 and human evaluators.

provement in terms of overall accuracy, as well as enhancing originality and conciseness in the JG task. We include examples of justifications generated by these models in Appendix A.3 for detailed illustration. Moreover, the high agreement between human evaluators and GPT-4 suggests that automated evaluation using GPT-4 is a more viable approach than similarity evaluation.

### 3.4 Comparison with NLI

Furthermore, we conduct an inference on SOUL test set using a widely used NLI model, namely the NLI-RoBERTa model[2], trained on the SNLI (Bowman et al., 2015) and MNLI (Williams et al., 2018) datasets, to demonstrate the focus of SOUL on subjective information rather than logical connections. As presented in Table 4, the NLI-RoBERTa model achieves an accuracy of only 55.02%, which is significantly lower compared to the RoBERTa model trained on the SOUL dataset. This outcome emphasizes the distinction between the objectives of SOUL and traditional NLI tasks. While they may share some similarities, the primary goal of SOUL is to extract and label subjective information, rather than establishing logical connections or entailment between different texts.

## 4 Conclusion

This paper introduces a novel task called Sentiment and Opinion Understanding of Language (SOUL), including two subtasks: review comprehension and justification generation. Our experimental results show that SOUL is a challenging task that demands a deep understanding of sentiment, with a performance gap of up to 27% when compared to human performance. Moreover, evaluations conducted with both human experts and GPT-4 demonstrate

| Model | Training Data | RC Accuracy |
|-------|---------------|-------------|
| NLI-RoBERTa | MNLI & SNLI | 55.02 |
| RoBERTa | SOUL | **75.49** |

Table 4: Comparison between SOUL and NLI tasks.

the weakness of SLMs in generating reasoning-based justifications while showcasing ChatGPT's powerful sentiment understanding ability. Nevertheless, there is still scope for enhancing the overall accuracy, originality, and conciseness of Chat-GPT's responses.

## Limitation

The newly proposed dataset SOUL utilizes customer reviews as the main source of constructing subjective statements. However, incorporating more opinionated texts, such as social media posts and dialogues, could potentially enable the assessment of models in a wider variety of text types. Also, SOUL currently features two tasks, including review comprehension and justification generation, to evaluate the model's sentiment understanding abilities. More task formats can be designed to comprehensively understand the model's capabilities and limitations.

## Acknowledgements

Y. Deng is supported by Alibaba Group through Alibaba-NTU Singapore Joint Research Institute (JRI), Nanyang Technological University, Singapore. Sinno J. Pan thanks for the support from HK Global STEM Professorship and the JC STEM Lab of Machine Learning and Symbolic Reasoning.

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

# A  Appendix

## A.1  Detailed Setup

We perform a grid search on the development set to find the best hyper-parameters for fine-tuning SLMs. Specifically, we search the learning rate among {1e-6, 5e-6, 1e-5, 5e-5, 1e-4}, batch size among {2, 4, 8}, and number of epochs among {4, 8}. For LLMs, we utilize their APIs to perform zero-shot inference.

## A.2  GPT-4 Prompt for Evaluation

We adopt the following prompt to evaluate the justification generated by different models:

```
Given a review and a statement,
along with their gold label and gold
justification, you will also receive
labels and multiple justifications
generated by different models. Your task
is to rate these generated justifications
based on the following metrics:

1. Correctness (scale of 1-3): Evaluate
whether the justification is sensible and
logical when compared to the gold label.
```

```
2. Alignment (scale of 1-3): Assess
whether the justification aligns with the
generated label.
3. Relevance (scale of 1-3): Determine
whether the justification is relevant to
the statement.
4. Conciseness (scale of 1-3): Evaluate
whether the justification is brief and
concise, without compromising clarity and
accuracy.
5. Originality (scale of 1-3): Assess
whether the justification demonstrates
innovation and uniqueness, rather than
simply copying the review and statement.

Please return your answers as a Python
dictionary, where the key is the model
name, and the value is a dictionary
containing the aforementioned metrics.
Please avoid returning any additional
text.
```

## A.3  Case Study

Table 5 presents examples of justifications generated by various models. In this particular sample, all models, except Flan-T5$_{\text{XXL}}$, have made the correct prediction. However, when it comes to justifications, both T5 and Flan-T5 have simply copied text from the review without any reasoning. On the other hand, ChatGPT has demonstrated a strong ability to understand sentiment by providing reasonable justifications based on the original review text, which led to the correct prediction.

**Review:** This is only related to the first movie by the name. The plot has nothing to do with the first and the whole movie stinks!!! I have no idea what they were thinking but this movie is so bad. Avoid this at all costs, the first movie in the series is acceptable as a slasher flick and so is the fourth but this one and the 3rd are rubbish!!
**Statement:** This movie is worse than the first one.
**Gold label:** true
**Gold justification:** The reviewer said that the first movie is still acceptable, but this one and 3rd are rubbish.

| | |
|---|---|
| T5 | **Label**: true 
 **Justification**: The reviewer said that the plot has nothing to do with the first and the whole movie stinks!!! |
| Flan-T5 | **Label**: true 
 **Justification**: The reviewer said that this one and the 3rd are rubbish!! |
| Flan-T5$_{XXL}$ | **Label**: false 
 **Justification**: The plot has nothing to do with the first and the whole movie stinks!!! I have no idea what they were thinking but this movie is so bad. Avoid this at all costs, the first movie in the series is acceptable as a slasher flick and so is the fourth but this one and the 3rd are rubbish!! |
| ChatGPT | **Label**: true 
 **Justification**: The reviewer explicitly states that "the whole movie stinks!!!" and that "this one and the 3rd are rubbish!!" in comparison to the first movie, which they found acceptable as a slasher flick. |

Table 5: Examples of justifications generated by various models, with colored text indicating the overlap with the original review text.