# OpenReview forum: "SOUL: Towards Sentiment and Opinion Understanding of Language"
_EMNLP/2023/Conference — EMNLP 2023 Main_

### Official Review · Reviewer_HN7A · 2023-08-01

**Soundness:** 3

**Excitement:**

4: Strong: This paper deepens the understanding of some phenomenon or lowers the barriers to an existing research direction.

**Paper Topic And Main Contributions:**

The paper presents a novel sentiment analysis task that goes beyond the traditional sentiment/aspect classification tasks. They aim to have the model assess statements on the sentiment of the text as well as generate justification for its decision on the statement label. They employ  two commonly-used sentiment classification datasets, the Yelp and the IMDP sets. They assign annotators to produce approx. 4 statements for each review along with their labels and justifications. The review dataset is around 36k reviews and the statements amount to 15k. They benchmark the dataset on trained SLMs and zero-shot settings with LLM.

The conclusion on their experiments is that all models show limited sentiment ability when it comes to justifying the sentiment label, specifically in their ability to generate reasoning behind the sentiment labeling. They also use GPT-4 prompts as an evaluation tool to their Sentiment Justification task. They show that GPT-4 performance in assessing the model's generated justification text is on par with the human evaluation.

**Questions For The Authors:**

A. What do you mean by overall accuracy in Justification Generation evaluation?
B. Can you give examples to the "not-given" label?
C. Why the review has statements such as "The reviewer doesn't like the movie" in the example given in the introduction. How is this statement different than just labeling the whole text as negative sentiment?

**Reasons To Accept:**

The main strengths of the paper are the following:

1. It presents a novel task that addresses the shortcomings of traditional polarity detection. Although sentiment classification benchmarks are significantly high and there are numerous pretrained sentiment detection models that illustrate very high performance, indirect expression of sentiment (e.g. irony) is still not accurately detected. The paper introduces a task that attempts to improve exactly this defect in  the sentiment classification task.

2. The justification task is a very good approach for reliable AI. The authors attempt to have the model not only decide on the sentiment label of the text but also justify why it makes the decision. Although the task seems quite challenging, but further expansion on the idea for sentiment analysis and other common NLP task can be very beneficial in minimizing the black box effect of the DL models.

3. The authors make use of the GPT prompting for evaluating the model justification generation output and they show the soundness of this method by comparison to human evaluators.
4. The paper is easy to read.

**Reasons To Reject:**

There are a number of points that need to be addressed:

1. The Justification Generation task results are confusing. The authors mention that ChatGPT accuracy is significantly  higher but Table 3 does not match their conclusion. Table 3 seems to show different results than the ones explained in section "Justification Generation".

2.  If ChatGPT is capable of producing the reasoning behind any sentiment classification label, then it is a solved problem. The ChatGPT justification in Table 4 is even better than the gold-standard. It would be more challenging to have justification of the sentiment label of the text as a whole rather than just the statement. The authors give a good example at the Introduction where indirect expression of emotion is particularly difficult to detect. Justification for the model's sentiment labeling to such an expression of sentiment would be a very good approach to the task of sentiment analysis.
3. The annotation schema rejects simple statements. It is not very clear the criteria for a simple statement. Also, an Inter Annotator Agreement would have substantiated the soundness of the data annotation.
4. A major problem with the annotation is the "not-given" label for the statements. There aren't any examples for statements with this label in the paper although there is a large number of statement with the not-given label in the training data. This label in particular can be quite confusing to the model specially with the "False" label.
5. In evaluating the generated text by the model, the authors report "overall accuracy" as well as BLEU, ROUGE and BERTScore. It is not clear how is this "overall accuracy" measured.
6. The text generation task is very difficult to evaluate by a non-human metric. It is not a straightforward text generation task, it is a justification of the model's decision. Human judgement would be a more reliable approach than the reported metrics.
7. The colors in Figure 2 are very hard to distinguish.


**Reproducibility:**

3: Could reproduce the results with some difficulty. The settings of parameters are underspecified or subjectively determined; the training/evaluation data are not widely available.

**Reviewer Confidence:**

5: Positive that my evaluation is correct. I read the paper very carefully and I am very familiar with related work.

**Typos Grammar Style And Presentation Improvements:**

Please correct grammar in line 071-072

---

> ### Author Rebuttal · Authors · 2023-08-28
>
> Thank you for your valuable and detailed feedback and suggestions. We have carefully considered each of your concerns and will make the necessary revisions to improve the clarity and comprehensiveness of our paper.
>
> > ***1. The Justification Generation task results are confusing***
>
> We apologize for any confusion caused and would like to clarify on the significant improvement of ChatGPT in justification generation. This improvement specifically relates to the accuracy of predicting statements and is compared to the performance achieved in the previous task, which solely predicts the statement without generating justification. Specifically, the accuracy has risen from 72.09% to 78.04%.
>
> > ***2. ChatGPT is capable of producing the reasoning & suggestion on more challenge task***
>
> Figure 2 demonstrates that ChatGPT's justifications currently lack conciseness and originality, highlighting the potential for further improvement in its performance.
>
> We greatly appreciate your valuable suggestion regarding the task format. However, this is a subset of our proposed task format, as we can still rephrase it into a statement and prompt the model to evaluate its correctness and generate a corresponding justification.
>
> > ***3.  Criteria for a simple statement***
>
> During the quality check, statements in SOUL are marked as simple, medium, or hard by annotators to determine their difficulty level. Reviews containing only simple statements are excluded to ensure an appropriate level of challenge.
>
> To evaluate the agreement between annotators, we employed Cohen's kappa coefficient, which yielded an impressive score of 0.9649. This high level of agreement reinforces the validity and reliability of the data annotation process.
>
> > ***4. Example of "not-given" label***
>
> Not-given refers to a statement that cannot be determined from the existing context. In the sample review shown in figure 1, an example of a not-given statement could be "The reviewer liked the raptors because they were terrifying." This statement cannot be concluded from the available information.
>
> On the other hand, a false statement can be directly inferred based on the context, such as "The reviewer liked the movie." We apologize for not providing an example of a "not-given" statement, and will include specific examples to clarify the definition of the "not-given" label.
>
> > ***5. how is this "overall accuracy" measured.***
>
> The overall accuracy is the same metric used in the RC task, representing the accuracy of all statements. The term "overall" is used to distinguish it from the class-wise F1 score. We apologize for not providing a clear definition of "overall accuracy" for the JG task, and we will include a formal definition in the revised version.
>
> > ***6. Human judgement would be a more reliable approach***
>
> We agree that evaluating the quality of generated justifications by humans is valuable, and we do have listed important criteria and adopted human evaluation, as shown in Section 3.3 and Figure 2. However, human evaluation can be time-consuming, costly, and subjective. Therefore, we adopted GPT-4 and found that it had high agreement with human evaluators, proving it to be a viable alternative. Still, we acknowledge the importance of human evaluation and encourage future users to conduct it, following our outlined criteria in Section 3.3, whenever resources permit.
>
> > ***7. The colors in Figure 2 are very hard to distinguish.***
>
> We apologize for the inconvenience caused by the figure's colors, and we will use more distinguishable and reader-friendly colors for Figure 2 in the revised version.
>
> > ***Question A. What do you mean by overall accuracy?***
>
> We have addressed this concern in #5.
>
> > ***Question B. Can you give examples to the "not-given" label?***
>
> We have addressed this concer in #4
>
> > ***Question C. Statement in example***
>
> While our annotation process encourages complex subjective statements, there still might be some relatively simple statements included. The example in Figure 1 is a randomly sampled instance and not cherry-picked.
>
> Actually, instead of just stating "The reviewer doesn't like the movie," our task format allow us to ask for specific details like "The reviewer finds the movie boring/terrifying/slow-paced/confusing," etc. Although they are all negative sentiments, not all of them may accurately reflect the review and could potentially be incorrect. That's one of our task advantages compared to traditional sentiment analysis tasks.
>
> > ***Typos, grammar***
>
> Thank you for pointing out the typographical error. We will correct them in the revised version of the paper.
>
> Thanks again for your detailed feedback, we believe that addressing these concerns will greatly enhance the quality and impact of our work.

---

### Official Review · Reviewer_TkL7 · 2023-08-03

**Soundness:** 3

**Excitement:**

3: Ambivalent: It has merits (e.g., it reports state-of-the-art results, the idea is nice), but there are key weaknesses (e.g., it describes incremental work), and it can significantly benefit from another round of revision. However, I won't object to accepting it if my co-reviewers champion it.

**Paper Topic And Main Contributions:**

This paper explores a novel task called sentiment and Opinion Understanding of Language (SOUL), which consists of review comprehension and justification generation.  Review comprehension focuses on verifying comprehension statements regarding an opinionated review text.  Justification generation focuses on generating justifications for such sentiment prediction. The authors annotate a SOUL dataset based on common review corpus Yelp and IMDb.  Experimental results on this dataset demonstrates that SOUL is a very challenging task.



**Questions For The Authors:**

a. No novelty method is proposed to solve SOUL.  The experiments focus on the small language models Roberta and Flan-T5.   What’s the performance of the SOTA MRC  and NLI methods on SOUL task?

b.Error analysis should be provided for figure out the key  challenges for SLM and LLM models in SOUL task.


**Reasons To Accept:**

This paper explores a novel task called Sentiment and Opinion Understanding of Language (SOUL), which consists of review comprehension and justification generation.

 It is an interesting topic and very useful in sentiment applications.

The authors provide a valuable SOUL dataset based on common review corpus Yelp and IMDb.


**Reasons To Reject:**

This paper doesn't provide a novelty method to solve SOUL.

The experiments focus on the small language models Roberta and Flan-T5.   The SOTA MRC methods should be evaluated.

Error analysis should be provided to figure out the gap between the existing models and human.



**Reproducibility:**

3: Could reproduce the results with some difficulty. The settings of parameters are underspecified or subjectively determined; the training/evaluation data are not widely available.

**Reviewer Confidence:**

5: Positive that my evaluation is correct. I read the paper very carefully and I am very familiar with related work.

---

> ### Author Rebuttal · Authors · 2023-08-28
>
> Thanks for your valuable feedback! We now address the concerns as below:
>
> > ***1. No novelty method is proposed***
>
> While proposing a novel method is indeed important, we would like to emphasize that the primary focus of our paper is to introduce the SOUL task. Our objective is to emphasize the importance of capturing the complexities of sentiment analysis through this task. We appreciate your input and will consider exploring novel methods in our future work.
>
> > ***2. SOTA NLI and MRC performance***
>
> Regarding the performance of SOTA NLI methods on the SOUL task, we conducted a quick inference using a popular NLI model, specifically the NLI-RoBERTa model trained on the MNLI and SNLI datasets (https://huggingface.co/cross-encoder/nli-roberta-base). We show its performance and the reported result of RoBERTa from Table 2 as below:
> | Model        | Training Data | Review Comprehension Accuracy |
> | ------------ | ------------- | ----------------------------- |
> | NLI-RoBERTa | MNLI & SNLI   | 55.02                         |
> | RoBERTa      | SOUL          | 75.49                         |
>
> The NLI-RoBERTa model only achieves an accuracy of 55.02%, which is significantly lower compared to RoBERTa model trained on SOUL dataset. This result highlights the distinction between the SOUL and traditional NLI tasks. While they share similarities, the main objective of SOUL is to extract and label subjective information rather than establishing logical connections or entailment between different texts.
>
> Regarding MRC, it differs significantly from the justification generation subtask in SOUL. MRC involves extracting answer spans from a passage based on a question, while justification generation in SOUL requires free-form generation, including common sense knowledge and detailed explanations. Additionally, the criterion of "originality", as discussed in section 3.3, is crucial in evaluating justifications in SOUL, emphasizing the need for unique and innovative responses. Therefore, directly adopting an MRC method may not be suitable.
>
> > ***3. Error analysis should be provided***
>
> Thank you for your feedback. We recognize the significance of error analysis in bridging the gap in the SOUL task. Following your suggestions, we have enhanced our analysis. Through a thorough error analysis, we have pinpointed two primary causes of mispredictions. Below are representative examples to illustrate these reasons:
>
> **Review**: Nice hotel, nice warm cookies!  I have stayed at this one twice and would recommend to out of towners.
>
> | Statement                                               | Label     | Prediction |
> | ------------------------------------------------------- | --------- | ------- |
> | The reviewer said the service was good.                 | Not-given | True    |
> | The reviewer said he would recommend it to his friends. | False     | True    |
>
> 1. **Superficial textual features**: The model tends to make incorrect predictions based on surface-level matching. In both statements, the model mistakenly predicts them as true because it simply match with the positive words in statement 1 and the "recommend" word in statement 2.
> 2. **Lack of common sense knowledge**: The model sometimes struggles to comprehend certain terms or phrases that require background knowledge or contextual understanding. In the second statement, the model fails to comprehend the term "out of towners" and makes an incorrect prediction.
>
> In our revised version, we will include a detailed error analysis section that delves into the key challenges faced by the SLM and LLM in the SOUL task.

---

### Official Review · Reviewer_Qqt5 · 2023-08-10

**Soundness:** 2

**Excitement:**

3: Ambivalent: It has merits (e.g., it reports state-of-the-art results, the idea is nice), but there are key weaknesses (e.g., it describes incremental work), and it can significantly benefit from another round of revision. However, I won't object to accepting it if my co-reviewers champion it.

**Paper Topic And Main Contributions:**

This paper describes a task of sentiment and opinion understanding of language (SOUL), which consists of two subtasks: review comprehension and justification generation.

**Reasons To Accept:**

It's a nice short paper that clearly explains the (seemingly new) task and the results of the experiments.

**Reasons To Reject:**

The authors did not explain how SOUL is different from explainable fact checking/textual entailment and how the statements are different from hypothesis generation for the textual entailment task. Unless they address these questions, I'm not convinced that the task they proposed is that new.

**Reproducibility:**

2: Would be hard pressed to reproduce the results. The contribution depends on data that are simply not available outside the author's institution or consortium; not enough details are provided.

**Reviewer Confidence:**

3: Pretty sure, but there's a chance I missed something. Although I have a good feel for this area in general, I did not carefully check the paper's details, e.g., the math, experimental design, or novelty.

---

> ### Author Rebuttal · Authors · 2023-08-28
>
> Thank you for your concern. We want to clarify that the proposed SOUL task differs fundamentally from textual entailment and explainable fact checking in its primary focus, i.e., **SOUL focuses on subjective sentiments and opinions while these existing studies focus on factual veracity**.
>
> More specifically, existing textual entailment or fact checking studies aims to judge the factual veracity or relation for a given text / text pair. However, the newly proposed SOUL task aims to encapsulate any sentiment-related issue through statement proposition and verification, thus breaking the constraints of traditional multiclass sentiment analysis tasks. Therefore, it requires a deep understanding of subjective information in the text to make correct predictions.
>
> To illustrate this distinction, we conducted a quick inference on the SOUL dataset using a RoBERTa model trained on the SNLI and MNLI datasets. Its accuracy is 55.02%. The performance is significantly lower compared to the RoBERTa result in Table 2, which is 75.49%. This discrepancy highlights the disparity between SOUL and traditional NLI tasks.
>
> Similarly, while explainable fact checking aims to establish the truth or falsehood of a factual claim, SOUL's justification generation subtask places greater emphasis on discerning the subtle variations in sentiment.
>
> In our revised manuscript, we will include a detailed comparison between SOUL and these tasks, highlighting the unique characteristics and contributions of SOUL, which can help to clarify the novelty of our proposed task and its distinction from existing tasks.

---

### Official Review · Reviewer_5t9T · 2023-08-11

**Soundness:** 3

**Excitement:**

4: Strong: This paper deepens the understanding of some phenomenon or lowers the barriers to an existing research direction.

**Paper Topic And Main Contributions:**

The paper topic is about Sentiment Analysis, and the authors contribution is two folded: (i) the annotated data set (with 15,028 statements) and (ii) the benchmark of new task SOUL (Sentiment and Opinion Understanding of Language) which provides two subtasks, the Review Comprehension and Justification Generation. The first one aims to validate statements based on a review text, whereas the latter one provides explanations for their sentiment predictions. SOUL uses SLM with complete training, including Roberta, T5 and Flan-T5 and extended their analysis to two representative LLMs, Flan-T5XXL and  ChatGPT. Both of them evaluated under zero-shot settings.

**Questions For The Authors:**


Q1: Why you have not indicated the results neither in abstract nor in the conclusion? This should be indicated into the final version of the paper.

**Reasons To Accept:**

The new task The results show that SOUL requires an in-depth understanding of sentiment. It is a difficult concept but also a promising strategy for sentiment analysis research. According to the results, Flan-T5 outperforms T5 among SLMs by 1.41% with the same model size. When it comes to performance, Flan-T5XXL surpasses ChatGPT while LLMs demonstrate effective zero-shot capability. When it comes to accuracy, ChatGPT performs better, with a notable improvement of roughly 6% in validating subjective statements. The results are encouraging, which led to the paper's acceptance.



**Reasons To Reject:**

N/A

**Reproducibility:**

5: Could easily reproduce the results.

**Reviewer Confidence:**

4: Quite sure. I tried to check the important points carefully. It's unlikely, though conceivable, that I missed something that should affect my ratings.

---

> ### Author Rebuttal · Authors · 2023-08-28
>
> Thank you for your comment and suggestion. We agree that it is important to provide a summary of the key findings in these sections to give readers a clear understanding of the results. In the revised version of the paper, we will ensure that the results are appropriately summarized in both the abstract and conclusion.

---

### Meta-Review · Area_Chair_PoFd · 2023-10-05

**Recommendation:** 3

**Metareview:**

This paper introduces the SOUL (Sentiment and Opinion Understanding of Language) task, which includes review comprehension and justification generation. It provides a dataset with 15,028 annotated statements and evaluates various models, including SLMs like Roberta, T5, and Flan-T5, as well as LLMs like Flan-T5XXL and ChatGPT, in zero-shot settings. The results highlight the challenge of justifying sentiment labels, with GPT-4 performing similarly to human evaluation in assessing generated justifications.

Reasons to accept:

Novel Task: Introduces the SOUL task that addresses shortcomings in sentiment analysis, focusing on detecting indirect sentiment expressions like irony.
Enhances AI model reliability by requiring justification for sentiment decisions.
Effective Evaluation: Uses GPT prompting for model justification evaluation, demonstrating its validity compared to human evaluation.
The paper is well-written and organised.

Reasons to reject:

Novel Method: The paper lacks a novel method to address the SOUL task. -Limited Model Evaluation: The experiments primarily focus on small language models (Roberta and Flan-T5), while state-of-the-art Machine Reading Comprehension (MRC) methods are not evaluated. -Missing Error Analysis: The paper lacks error analysis to identify gaps between existing models and human performance. -Confusing Results: The results for the Justification Generation task are inconsistent.
Ambiguity in Annotation Criteria and lack of Inter Annotator Agreement for data annotation. -The paper reports an "overall accuracy" metric without explaining how it's measured, and relies on non-human metrics like BLEU, ROUGE, and BERTScore for text generation evaluation, which may not be suitable for this task.

---

### Decision · Program_Chairs · 2023-10-07

**Decision:**

Accept-Main

**Comment:**

This paper introduces the SOUL (Sentiment and Opinion Understanding of Language) task, which includes review comprehension and justification generation. It provides a dataset with 15,028 annotated statements and evaluates various models, including SLMs like Roberta, T5, and Flan-T5, as well as LLMs like Flan-T5XXL and ChatGPT, in zero-shot settings. The results highlight the challenge of justifying sentiment labels, with GPT-4 performing similarly to human evaluation in assessing generated justifications.

Reasons to accept:

Novel Task: Introduces the SOUL task that addresses shortcomings in sentiment analysis, focusing on detecting indirect sentiment expressions like irony.
Enhances AI model reliability by requiring justification for sentiment decisions.
Effective Evaluation: Uses GPT prompting for model justification evaluation, demonstrating its validity compared to human evaluation.
The paper is well-written and organised.

Reasons to reject:

Novel Method: The paper lacks a novel method to address the SOUL task. -Limited Model Evaluation: The experiments primarily focus on small language models (Roberta and Flan-T5), while state-of-the-art Machine Reading Comprehension (MRC) methods are not evaluated. -Missing Error Analysis: The paper lacks error analysis to identify gaps between existing models and human performance. -Confusing Results: The results for the Justification Generation task are inconsistent.
Ambiguity in Annotation Criteria and lack of Inter Annotator Agreement for data annotation. -The paper reports an "overall accuracy" metric without explaining how it's measured, and relies on non-human metrics like BLEU, ROUGE, and BERTScore for text generation evaluation, which may not be suitable for this task.